# Is the Coming Out of an LGBTQIA+ Child a Death-like Event for Italian Parents?

Nicola Biancotto [1,*], Gianmarco Biancalani [1], Lucia Ronconi [2] and Ines Testoni [1]

[1] Department of Philosophy, Sociology, Education and Applied Psychology, University of Padua, 35122 Padua, Italy; gianmarco.biancalani@unipd.it (G.B.); ines.testoni@unipd.it (I.T.)

[2] Computer and Statistical Services, Multifunctional Pole of Psychology, University of Padua, 35122 Padua, Italy; l.ronconi@unipd.it

[*] Correspondence: nicolabiancotto1@gmail.com

**Abstract:** Parents of LGBTQIA+ individuals often report experiencing an affective state similar to grief after their children's coming out. The current study explores whether this experience resembles that of people who have recently lost someone close. Furthermore, we tested whether the parents' alexythimic traits are associated with their grief-like experience. In a sample of 194 parents who experienced their children's coming out, we administered the Integration of Stressful Life Events Scale (ISLES), the Social Meaning In Life Events Scale (SMILES), and the Toronto Alexithymia Scale (TAS-20). The results showed no significant differences in the mean scores of ISLES and SMILES between the present and bereaved samples by their creators. In addition, in the present sample, lower ISLES and SMILES scores were associated with higher alexithymic traits. Overall, these findings suggest a resemblance between the experience of parents following their children's coming out and that of bereaved individuals. Therefore, they could inform on how to assist parents in coming to terms with the coming out of an LGBTQIA+ child.

**Keywords:** coming out; parents; stressful experience; LGBTQIA+ child; loss; social meaning-making





## 1. Introduction

Coming out is often an arduous process, as announcing that they do not fit into heteronormative and cis-normative societal expectations is likely to induce stress, fear, and discomfort for individuals identifying themselves as lesbian, gay, bisexual, transgender, intersex, asexual, or any other gender identity or sexual orientation that is not cisgender or heterosexual (LGBTQIA+) (Rosati et al. 2020). The process of coming out is crucial for the happiness and wellbeing of LGBTQIA+ individuals, although its repercussions often affect their parents as well. Some parents may go through a coming out process paralleling their child's, as they must develop new identities, crafting new personal narratives as parents of an LGBTQIA+ child (Carbone et al. 2022), but still feeling the need to conceal their new identity (Goodrich 2009). This behavioral pattern also appears to extend to parents who immediately show acceptance and support following their child's disclosure (Trussell 2017). The latter also seem to experience many negative emotions, such as loss, fear, hurt, denial, self-blame, shame, guilt, or even despair (Goodrich 2009).

Multiple studies suggest that parental responses may change and evolve as time passes (Broad 2011; Fields 2001; Phillips and Ancis 2008). The adjustment following the children's coming out tends to progress through three broad phases, with each having emotional, cognitive, behavioral, moral, and spiritual aspects. Initially, emotionally driven responses dominate the parents' first reaction. In the intermediate phase, parents often emphasize cognitive and behavioral strategies while attending to the emotional issues related to external people's perspectives and their possible judgment. The third and final adjustment phase is ideally characterized by the resolution of moral and spiritual issues, allowing parents to fulfill their emotional, social, and moral needs and duties by being

able to love their children unconditionally (Broad 2011; Phillips and Ancis 2008; Testoni and Pinducciu 2019; Trussell 2017). However, the possibility of a positive outcome does not invalidate the parents' negative emotions and experiences following their children's coming out. An experience often shared by the parents of LGBTQIA+ children following their coming out is an undefined feeling of loss and grief. Qualitative research has long collected testimonies from parents expressing these sentiments. For example, Fields in 2001 interviewed several individuals whose children had come out who described their experience as: "I am grieving here. I feel as if I lost my son." and "I guess it's like any other crisis- cancer, a car accident. You need time to recover" (Fields 2001). More recent qualitative studies delineated similar findings: participants reported deep sadness, loss, and grief similar to the feelings caused by the death of someone close (Horn and Wong 2017; Katz-Wise et al. 2016; Saltzburg 2004; Testoni and Pinducciu 2019). However, to our knowledge, no quantitative study has investigated this issue. Therefore, an important goal of this research is to corroborate the findings of previous qualitative studies via an exploratory study using quantitative data.

Regarding bereaved individuals, two extensively studied factors are the integration of the loss event and their meaning-making efforts (Bellet et al. 2019; Testoni et al. 2020). Bereaved individuals' experiences resemble those exhibited by individuals who experienced other stressful life events (Graci et al. 2018). Stressful life events are generally defined as undesirable, unscheduled, nonnormative, uncontrollable, discrete, or observable events with a clear onset and offset that usually indicate significant life changes that have significant negative consequences for physical and psychological wellbeing (Carlson 2014). Bereavement often resembles a stressful life event, especially in a family context, and may require a long time to be processed while still being vulnerable to complications that could endanger the bereaved person's self-esteem and hope for the future. Furthermore, stressful events are situations experienced by individuals as a problem beyond their ability to manage, which hinders their wellbeing and daily functioning (Ouagazzal et al. 2021).

The integration process of a stressful life event is usually described as following two possible pathways. Usually, narratives describing stressful life experiences would be assimilated within one's preexisting self-narratives and models of the world created according to previous life experiences (Holland et al. 2010). However, when this process is impaired, as it often happens when experiencing traumatic or even just stressful life events, an individual may not be able to fully make sense of their experience, making it necessary to reconsider and possibly alter one's internal models to accommodate the new discrepant information about themselves and the world around them (Holland et al. 2010). In addition, one of the most crucial steps in coming to terms with stressful or traumatic experiences is to try to make meaning of what happened, to find a reason why the event may fit the narrative of one's life. The process of crafting meaningful narratives of a stressful experience is considered crucial to allow an individual to live and function in daily life with the memories of what happened to them. Meaning-making is one possible mechanism that may allow individuals to recover from a stressful event because it theoretically facilitates both coping and the resolution of these experiences in beneficial ways (Graci et al. 2018). Although the integration process of a stressful life event and subsequent meaning-making requires considerable individual and intrapersonal work, it ultimately requires survivors to recruit social validation for their changed identities (Neimeyer et al. 2021). This idea is consistent with long-accepted scientific research that suggests that social support is directly associated with better physical and mental health, general wellbeing, and routinely has a protective effect against the impact of adverse and stressful life events (Thoits 2013). Indeed, the meaning that individuals make of a stressful life-changing experience is influenced by how others perceive and react to the event and the following distress. From a social constructionist perspective (Neimeyer et al. 2021), dealing and coping with the aftermath of a stressful life event takes place in a complex social environment. In a broader context, the integration and meaning-making processes are affected by the communities' views on the specific events and the societal norms policing how an individual expresses, acts upon,

and copes with distress caused by the event. These social aspects can either encourage the individuals' meaning-making attempts by validating and honoring their distress or inhibit such attempts (Bellet et al. 2019; Testoni et al. 2020). Therefore, in the context of meaning-making, social interaction may be validating or invalidating. In the case of a positive outcome, the people surrounding the individual may support their attempts to make sense of the experience and how it fits in the broader narrative of their life or hinders their integration and subsequent recovery process that are necessary for healthy functioning (Bellet et al. 2019; Hasson-Ohayon et al. 2017; Holland et al. 2010).

*The Present Study*

The present study aims to explore whether the integration and meaning-making processes experienced by parents in the aftermath of their children coming out may be similar to those of individuals dealing with the death of someone close. To reach this goal, we compared our sample of collected data with the data of published studies in which the Integration of Stressful Life Experiences Scale (ISLES) and the Social Meaning in Life Events Scale (SMILES) were first put forward. Furthermore, this research explores the association among the parents' integration and their meaning-making processes and alexithymic levels.

## 2. Materials and Methods

### 2.1. Participants

The inclusion criteria to participate in this study included having resided in Italy consistently for most of one's life and being the parent of an individual belonging to the LGBTQIA+ community who has come out. The individuals comprising the sample included 152 women, 40 men, and 2 who identified as "other". Their age ranged from 32 to 78, with a mean of 58.62 (SD = 7.75). Most participants resided in the northern regions of Italy. It is notable that, although people seem to be equally distributed between cities of different sizes, people coming from metropolises represented only 9%. People composing the sample tended to be politically more left-leaning than the average population, and most of them considered themselves Christian but with different degrees of devotion. Given the difficulties of recruiting parents of LGBTQIA+ individuals willing to participate in this research, we have included parents of lesbian, gay, bisexual, transgender, queer, intersex, and asexual individuals, even though we acknowledge that these groups may differ from each other on characteristics that may affect their parents' process of accepting them. A more thorough description of the sample can be found in Table 1.

**Table 1.** Demographic characteristics of participants and their children (N = 194).

| Variable | Mean (SD)/N (%) | Variable | Mean (SD)/N (%) |
|---|---|---|---|
| Age | 58.62 (7.75) | Religion | |
| Gender | | Christian | 137 (71%) |
| Female | 152 (78%) | Atheist | 27 (14%) |
| Male | 40 (21%) | Agnostic | 23 (12%) |
| Other | 2 (1%) | Other | 7 (4%) |
| Educational level | | Intensity of religious belief | |
| Middle school | 17 (9%) | Low level | 55 (28%) |
| High school | 82 (42%) | Medium level | 60 (31%) |
| Graduation | 94 (48.5%) | High level | 39 (20%) |
| Missing | 1 (0.5%) | Number of children | |
| Marital status | | One | 49 (25%) |
| Unmarried | 2 (1%) | Two | 106 (55%) |
| Married/Cohabitant | 137 (71%) | Three or more | 39 (20%) |
| Separated | 16 (8%) | Child's gender identity | |
| Divorced | 23 (12%) | Woman | 59 (30%) |

**Table 1.** *Cont.*

| Variable | Mean (SD)/N (%) | Variable | Mean (SD)/N (%) |
|---|---|---|---|
| Widowed | 16 (8%) | Trans-woman | 15 (8%) |
| Geographic area | | Not-binary | 14 (7%) |
| Northern Italy | 127 (65.5%) | Man | 84 (43%) |
| Southern Italy | 67 (34.5%) | Trans-Man | 19 (10%) |
| City size | | Missing | 3 (1.5%) |
| Village | 45 (23%) | Child's sexual orientation | |
| Small town | 58 (30%) | Bisexual | 30 (15.5%) |
| Medium town | 45 (23%) | Heterosexual | 15 (8%) |
| Big town | 29 (15%) | Gay/Lesbian | 137 (71%) |
| Metropolis city | 17 (9%) | Pansexual | 7 (4%) |
| Employment Status | | Other | 5 (3%) |
| Worker | 120 (62%) | Time since the child's coming out | |
| Retired | 52 (27%) | less than one year | 11 (6%) |
| Housewife | 17 (9%) | 1–2 years | 24 (12%) |
| Other | 5 (3%) | 2–3 years | 23 (12%) |
| Income | | 3–4 years | 23 (12%) |
| <15,000 | 22 (11%) | 4–5 years | 20 (10%) |
| 15,000–25,000 | 55 (28%) | 5–10 years | 43 (22%) |
| 25,000–50,000 | 83 (43%) | more than 10 years | 50 (26%) |
| 50,000–100,000 | 26 (13%) | Suspicion before coming out | |
| >100,000 | 8 (4%) | No | 103 (53%) |
| Political orientation | | Yes | 91 (47%) |
| Right-wing | 4 (2%) | Loved ones who were already part of the LGBTQ+ community | |
| Centre-Right | 10 (5%) | | |
| Centre | 14 (7%) | No | 106 (55%) |
| Centre-Left | 57 (29%) | Yes | 88 (45%) |
| Left-wing | 94 (48.5%) | | |
| Missing | 15 (8%) | | |

## 2.2. Data Collection

Participants were recruited online on social media platforms such as Facebook, Whatsapp, Instagram, and Telegram via direct messaging or posting the questionnaire link on groups devoted to LGBTQIA+ topics. The recruitment message included a statement explaining that the general purpose of the research was to explore the experience of parents of LGBTQIA+ individuals who have come out. After assuring the anonymity of the responses, the participants received a link to the online questionnaire, which took around twenty minutes to complete. Both the recruiting message and the questionnaire were presented only in Italian.

Moreover, several local non-profit LGBTQIA+ associations and social media personalities helped increase the number of complete questionnaires by circulating the link to a larger audience. Specifically, members of these LGBTQIA+ non-profit associations forwarded the message to their parents and people they knew who fit the inclusion criteria following a snowball model. In addition, these associations sent the link to the questionnaire to their signed-up mailing list.

## 2.3. Measures

### 2.3.1. Socio-Demographical Characteristics

The questionnaire included an initial page to collect personal and demographic information, such as the participant's gender, age, political inclinations (by choosing among "Far left", "Left", "Center-left", "Center", "Center-right", "Right", and "Far Right"), their child's sexual orientation (by choosing among "Straight", "Lesbian", "Gay", "Bisexual", "Asexual", and "Other"), and gender identity (by choosing among "Cis Man", "Cis Woman", "Trans Man", "Trans Woman", "Non Binary", and "Other"), before including the following four measures.

### 2.3.2. Integration of Stressful Life Experiences Scale

The Integration of Stressful Life Experiences Scale (ISLES; Holland et al. 2010) is a 16-item 5-point Likert scale (1-strongly agree; 5-strongly disagree) that measures the degree to which a stressful life experience has been adaptively incorporated into one's broader life story to promote a sense of internal coherence and foster a secure and hopeful view of the future. This research used the Italian-validated version of the measure (Neimeyer et al. 2021). In the present study, the scale had good reliability (Cronbach's alpha = 0.93). Item example: "My beliefs and values are less clear since this event".

### 2.3.3. Social Meaning in Life Events Scale

Social Meaning in Life Events Scale (SMILES; Bellet et al. 2019) is a 24-item 5-point Likert scale (1-strongly disagree; 5-strongly agree) that evaluates the degree to which social interactions ease or hinder an individual's ability to make sense of significant stress factors, trauma, or loss. The scale has a two-factor structure yielding two independent subscales: Social Invalidation (the extent to which the people around them invalidated a mourner's efforts to make meaning) (Item example: "I worry that if I shared too much about this event, people might see me differently"), and Social Validation (the extent to which the people around the mourner validated their meaning-making) (Item example: "Opening up about what happened has helped bring resolution to the situation."). In the present study, Social Invalidation had good reliability (Cronbach's alpha = 0.95) and Social Validation also had good reliability (Cronbach's alpha = 0.84). This research used the Italian-validated version of the measure (Testoni et al. 2020).

### 2.3.4. Toronto Alexithymia Scale

The Toronto Alexithymia Scale (TAS-20; Bagby et al. 1994) is a 20-item 5-point Likert scale (1-strongly disagree; 5-strongly agree) to assess the level of alexithymia, defined as the inability to recognize or describe one's own emotions. The scale has a three-factor structure yielding three independent subscales: Difficulty Identifying Feelings, Difficulty Describing Feelings, and Externally-Oriented Thinking. This research used the validated Italian version of the measure (Bressi et al. 1996). In the present study, the scale had good reliability (Cronbach's alpha = 0.93). Item example: "I am often confused about what emotion I am feeling".

### 2.4. Data Analysis

First, we compared the present mean ISLES scores with the mean scores obtained by the bereaved sample originally to validate the ISLES by Holland et al. (2010), using an independent sample *t*-test.

Second, we compared the present mean SMILES scores with both the mean scores obtained by the bereaved sample originally by Bellet et al. (2019) to validate the SMILES, using an independent sample *t*-test.

Afterwards, we examined the bivariate correlations between ISLES and SMILES with the socio-demographic characteristics of the participants and with TAS-20, using the Pearson correlation coefficient for continuous variables (i.e., parent's age); the Sperman rank correlation coefficient for ordinal variables (for example political orientation); and the *t*-test or analysis of variance for categorical variables with two or more categories.

Analyses were carried out using IBM SPSS Statistic 28 for descriptive, reliability, *t*-test, correlations, and mediation analysis.

## 3. Results

### 3.1. Comparison of ISLES with Bereaved Sample by Holland et al. (2010)

The ISLES scores of the participants whose experience of their children's coming out were compared to the ones found by Holland and colleagues in 2010. No statistically significant difference (t = 1.85 df = 404 *p* = 0.065) emerged when comparing the scores of the two groups that lived with the stressful experience within three years (Table 2).

**Table 2.** Comparison of ISLES and SMILES with bereaved samples.

| Measure | Research Sample | Comparison Sample | Test *t* | | |
|---|---|---|---|---|---|
| | **M (SD)** | **M (SD)** | **t** | **df** | ***p*-Value** |
| ISLES [1a] | 67.84 (12.79) | 61.66 (12.72) | 5.41 | 540 | <0.001 |
| ISLES [1b] | 65.07 (14.66) | 61.66 (12.72) | 1.85 | 404 | 0.065 |
| SMILES Social Invalidation [2a] | 2.42 (1.06) | 2.33 (0.70) | 1.19 | 540 | 0.236 |
| SMILES Social Invalidation [2b] | 2.49 (1.00) | 2.33 (0.70) | 1.51 | 404 | 0.133 |
| SMILES Social Validation [2a] | 3.31 (0.88) | 3.15 (0.65) | 2.40 | 540 | 0.017 |
| SMILES Social Validation [2b] | 3.23 (0.83) | 3.15 (0.65) | 0.83 | 404 | 0.406 |

[1a] Research sample = parents with experience of children's coming out (N = 194); Comparison sample = bereaved people of Holland et al. (2010) study. [1b] Research sample = parents with experience of recent children's coming out (N = 58); Comparison sample = bereaved people of Holland et al. (2010) study. [2a] Research sample = parents with experience of children's coming out (N = 194); Comparison sample = bereaved people of Bellet et al. (2019) study. [2b] Research sample = parents with experience of recent children's coming out (N = 58); Comparison sample = bereaved people of Bellet et al. (2019) study.

### 3.2. Comparison of SMILES with Bereaved Sample by Bellet et al. (2019)

Regarding the Social Validation factor of SMILES, no significant (t = 0.83, df = 404, and $p = 0.406$) difference appeared when comparing the data only from those parents whose experience of their children's coming out is more recent (within three years) with the data of the bereaved sample (loss within three years). A different pattern appeared regarding the Social Invalidation subscale. The Social Invalidation values of the entire sample group (M = 2.42, SD= 1.06) were not significantly different (t = 1.19, df = 540, and $p = 0.236$) from those of the bereaved sample (M = 2.33, SD = 0.70). The same results (t = 1.51, df = 404, and $p = 0.133$) were found when considering only the data from those parents whose children came out only in the previous three years (M = 2.49, SD = 1.00) (Table 2).

### 3.3. Correlations of ISLES and SMILES with Socio-Demographical Characteristics

Each socio-demographical characteristic, see Table 3, was examined to check for possible correlations between it and the ISLES and SMILE scores. A negative correlation between the parent's age and their ISLES scores was found (r = −0.15 $p = 0.039$), showing that the younger the parent, the higher the level of integration. We also found differences on the SMILES Social Invalidation score based on the parent's suspicion that the child was not heterosexual before they came out. Indeed, those parents who did not suspect their children of being a part of the LGBTQ+ community (M = 2.56 DS = 1.12) scored higher than those who had already expected it (M = 2.26, SD = 0.97; t = −1.96, df = 192, and $p = 0.051$), showing how they experienced higher levels of Social Invalidation. A significant positive correlation between the left-wing political orientation of the participants and their ISLES scores was found (rho = 0.20, $p = 0.009$), showing that left-leaning parents display better integration. An ANOVA also showed differences based on the child's gender identity in the SMILES Social Invalidation score ($F_{(4,186)} = 3.81$, $p = 0.005$): Parents whose children's gender identity is "Trans Woman" have showed higher levels of Social Invalidation (M = 3.38, SD = 0.96) than those who declared their children's gender identity to be "Woman" (M = 2.26, DS = 0.99) or "Man" (M = 2.35 DS = 1.00). No other significant relation emerged when investigating any of the remaining socio-demographical variables of each participant and their ISLES and SMILE scores.

**Table 3.** Correlation of study variables with socio-demographical characteristics.

| Variable | ISLES | | SMILES (SI) | | SMILES (SV) | |
|---|---|---|---|---|---|---|
| | Statistic [1] | *p*-Value | Statistic [1] | *p*-Value | Statistic [1] | *p*-Value |
| Age | r = −0.15 | 0.039 | r = 0.04 | 0.604 | r = 0.01 | 0.835 |
| Gender (Males vs. Females) | t = 0.59 | 0.554 | t = −1.14 | 0.255 | t = 0.33 | 0.741 |
| Education (No Graduation vs. Graduation) | t = 0.56 | 0.576 | t = 0.02 | 0.981 | t = 0.51 | 0.607 |
| Marital status (Cohabitant vs. Other) | t = −0.73 | 0.466 | t = 0.56 | 0.576 | t = 0.25 | 0.803 |
| Geographic area (Northern vs. Southern) | t = 0.23 | 0.815 | t = 0.31 | 0.756 | t = −0.07 | 0.947 |
| City size | rho = −0.06 | 0.440 | rho = −0.03 | 0.717 | rho = −0.02 | 0.767 |
| Employment Status | F = 0.24 | 0.868 | F = 0.62 | 0.604 | F = 0.45 | 0.718 |
| Income | rho = −0.01 | 0.880 | rho = −0.08 | 0.263 | rho = 0.07 | 0.348 |
| Political orientation | rho = 0.20 | 0.009 | rho = −0.07 | 0.338 | rho = 0.04 | 0.552 |
| Religion | F = 0.08 | 0.920 | F = 1.91 | 0.150 | F = 0.05 | 0.950 |
| Intensity of religious belief | rho = −0.00 | 0.952 | rho = 0.01 | 0.854 | rho = 0.02 | 0.755 |
| Number of children | rho = −0.05 | 0.475 | rho = −0.00 | 0.967 | rho = 0.13 | 0.072 |
| Child's gender identity | F = 1.89 | 0.113 | F = 3.81 | 0.005 | F = 1.61 | 0.172 |
| Child's sexual orientation | F = 0.57 | 0.635 | F = 1.03 | 0.379 | F = 0.67 | 0.568 |
| Time since coming out | rho = 0.10 | 0.179 | rho = −0.04 | 0.613 | rho = 0.04 | 0.562 |
| Suspicion before coming out | t = 1.40 | 0.164 | t = −1.96 | 0.051 | t = −0.60 | 0.549 |
| Already close to LGBTQIA+ individuals | t = 1.68 | 0.094 | t = −0.10 | 0.916 | t = 0.83 | 0.406 |

[1] Statistics used are: Pearson r coefficient for continuous variables, Spearman rho coefficient for ordinal variables, test t or one-way anova for dummy or multi-categories variables.

### 3.4. Correlations of ISLES and SMILES with TAS-20

As shown in Table 4, a strong positive correlation was observed between the SMILES Social Invalidation and the TAS-20 scores (r with values between 0.54 and 0.66), showing that the higher the level of alexithymic traits, the higher the levels of perceived social invalidation. There was also a moderate negative correlation between the SMILES Social Validation and the TAS-20 scores (r with values between −0.33 and −0.45), showing that the lower the level of alexithymic traits, the higher the levels of perceived social validation. A small negative correlation was observed between the ISLES and TAS-20 total scores (r = −0.16).

**Table 4.** Correlations of ISLES and SMILES with TAS-20.

| | TAS-20 Total | TAS-20 Factor1 | TAS-20 Factor2 | TAS-20 Factor3 |
|---|---|---|---|---|
| ISLES | −0.16 * | −0.14 | −0.20 ** | −0.09 |
| SMILES Social Invalidation | 0.66 ** | 0.63 ** | 0.61 ** | 0.54 ** |
| SMILES Social Validation | −0.43 ** | −0.34 ** | −0.38 ** | −0.45 ** |

\* $p < 0.05$; \*\* $p < 0.01$.

## 4. Discussion

The results of this study detail how the parents of LGBTQIA+ children labor to achieve the integration of the child's coming out into their personal life narratives similarly to those who grieve a recent loss. The comparison of the integration of stressful life events and social meaning-making levels between the present sample and the bereaved samples by Holland et al. (2010) and Bellet et al. (2019) indicates that a child's coming out may be, for their parents, a stressful life event similar to the death of a loved one. Furthermore, when considering participants who experienced disclosure less recently, the data show that the integration and social meaning-making efforts seem to allow them to achieve greater wellbeing over time. This is in line with the scientific literature that theorizes that parents slowly integrate and make sense of this stressful experience and perceived loss as they grow past the negative feelings associated with it (Broad 2011; Carbone et al. 2022; Goodrich 2009; Trussell 2017), as it often happens when dealing with the grief caused by the death

of someone close. Therefore, considering the results of this investigation, it seems that the aftermaths of both the disclosure event and the loss of a dear person follow similar patterns.

The time aspect was also investigated by comparing the entire sample group to those who only had their children come out in the previous three years, to illustrate the progression over time of the integration and acceptance of this stressful event. Interestingly, the social validation factor for social meaning-making appears to follow the expected pattern of improvement over time. However, after many years, participants still score similarly to the bereaved sample on the social invalidation score of social meaning-making. The social invalidation perceived by parents does not appear to decrease with time; this may be due to the stigma still surrounding the LGBTQIA+ status. It may be that socially widespread homonegativity and trans-negativity, both internalized and made apparent, are the cause of the continuing sense of invalidation parents perceive even many years later (Arayasirikul et al. 2022; Broad 2011; Norton and Herek 2013; Primo et al. 2020; Puckett et al. 2015). However, it is important to acknowledge that, due to the cross-sectional nature of this study, this data does not allow us to determine a causal relation between time and parental response, but only suggests an interaction to be further studied in future research.

Several interesting results also emerged regarding the correlation between each participant's socio-demographical characteristics and their success in their integration and social meaning-making processes. The higher age of the participants was associated with consistently lower scores, suggesting that older parents find it more challenging to integrate, accept, and make meaning of the disclosure of their children's minority status. The notion that older cohorts are more likely to hold stronger homonegative and trans-negative opinions, attitudes, beliefs, and stereotypes could explain such findings and indicate that it would be wise to consider older parents as more at risk for struggling to come to terms with their children coming out (Norton and Herek 2013).

In addition, parents who did not suspect that their child belonged to the LGBTQIA+ community seem to experience higher social invalidation levels than those who already knew or even just suspected the truth before their children's active disclosure. Based on these data, it could be theorized that, even if only based on suspicion, this knowledge may have allowed parents to begin the integration and meaning-making process beforehand or even gather the social and personal resources needed to facilitate the acceptance of this event.

Political orientation also related with the integration of this stressful event, as a higher integration was achieved and associated with parents who were left-wing leaning. This may be because politically left-leaning individuals are more likely to express concern about social justice and equal rights (Sakallı et al. 2019). Therefore, these results suggest that assistance in overcoming anti-rights views on LGBTQIA+ may represent a possible avenue of investigation for devising new methods of supporting parents in their integration and acceptance journeys.

Similarly, parents who have an LGBTQIA+ individual close to the family seem to have a better integration of the disclosure event than those who do not. A possible reason for this difference may be twofold. Parents who are friends with LGBTQIA+ people should be less likely to hold strong homonegative and transnegative opinions. Moreover, these parents may have at least some positive representation and information on what their child's life could be like, as well as have someone who is familiar and intimate with similar situations that they can talk to and guide them. Previous studies corroborate such an explanation showing the importance of dialogue and information when integrating being a parent to an LGBTQIA+ child into one's old identity (Broad 2011; Phillips and Ancis 2008).

The results of the current research indicate that the child's gender identity significantly affects both their parents' integration process and the invalidation perceived in the social meaning-making journey as shown by the low scores of parents of transwomen. The reason why parents seem to struggle most when having to accept a transfeminine child may find its roots in the fact that transwomen are often the most discriminated group due to the

intersection of transnegative and misogynistic attitudes that are widespread in patriarchal societies (Arayasirikul et al. 2022; Primo et al. 2020; Testoni and Pinducciu 2019).

Interestingly, many of the socio-demographical characteristics we expected to correlate with how parents manage the coming out of a child do not seem to have significant relations with this process. Indeed, while parents may differ in aspects such as religion, geographical origin, and the number of children they have, the data from the present suggests that this does not imply a difference in their integration of stressful events and the meaning-making process.

The current study also shows that alexithymia is related to the integration and meaning-making process that follows the disclosure of a child's minority status. People with higher alexithymia levels tend to perceive significantly stronger Social Invalidation and slightly weaker Social Validation, which may be associated with a stronger struggle with integrating the stressful event. Arguably, in parents with high alexithymic levels, struggling to handle, understand, and act upon one's feelings coherently may cause them to find it more challenging to complete the integration and meaning-making process following a stressful event (Goerlich 2018).

## 5. Conclusions

The primary purpose of this study was to explore whether the coming out of LGBTQIA+ children may represent a stressful event similar to grief for their parents. In so doing, we also aimed to corroborate the findings of previous qualitative studies reporting on the experience of parents of LGBTQIA+ children using quantitative methods. The results showed that parents whose children came out recently have similar levels of integration and meaning-making to those of bereaved individuals. In light of such results, it can be argued that a child disclosing their sexual orientation and gender identity minority status may represent a stressful and destabilizing event, not dissimilar to the loss of a dear person. Therefore, the techniques used to help bereaved individuals come to terms with their grief may also benefit the parents' integration and social meaning-making processes following a child coming out.

In addition, according to the results, high alexithymic traits correlate with lower levels of integration and successful social meaning-making. Therefore, practitioners may also direct their goals towards improving alexithymia-related issues of parents trying to come to terms with their children's coming out.

## 6. Limitations

The present sample presents an important limitation. The fact that a child's minority status may be a sensitive subject due to the still common homonegative and transnegative attitudes in Italy (Testoni and Pinducciu 2019) must be considered when analyzing the difficulties in recruiting participants for this study. It stands to reason that parents struggling to accept their LGBTQIA+ child would be far less likely to participate in the experiment. Therefore, involuntarily excluding parents with the most difficulties made our sample less representative, which may have influenced the results of this study, causing them to portray a better and healthier situation than what is accurate. Furthermore, it must be acknowledged that our sample could not be fully representative of the Italian population. For example, the participants' gender was not equally balanced, which may influence the results obtained.

Another limitation of this study is that the data collected by the present authors was compared to the data collected by previous researchers using the same measures, an issue that will be addressed in the following section.

## 7. Future Research

Given that the difficulty for parents of LGBTQIA+ children emerged with a non-completely random sample in the present study, further research should be conducted with a more representative sample, which may show even more pronounced results. In addition,

future research should replicate the present findings by having bereaved individuals and parents of LGBTQIA+ children within the same experimental design. Similarly, this topic should also be examined in other countries and cultural contexts to see if similar findings can be replicated.

To expand on the relationship between alexithymia and how parents react to their children's disclosure, future researchers could test whether encouraging struggling individuals to develop emotional skills would help them come to terms with their new reality.

Furthermore, longitudinal studies investigating this topic would be invaluable to verify our findings on how parents cope and work through the process of social meaning and integration and more fully accept their LGBTQIA+ children.

**Author Contributions:** Conceptualization, N.B.; methodology, N.B. and G.B.; software, L.R.; formal analysis, L.R.; investigation, N.B.; data curation, L.R.; writing—original draft preparation, N.B.; writing—review and editing, N.B. and L.R.; visualization N.B. and G.B.; supervision, I.T. and G.B.; project administration, I.T.; funding acquisition, I.T. All authors have read and agreed to the published version of the manuscript.

**Funding:** This research received no specific grant or funding from any institution in the public, commercial, or not-for-profit sectors.

**Institutional Review Board Statement:** The study was conducted in accordance with the Declaration of Helsinki, and approved by the Institutional Ethics Committee of University of Padova (No. 10E8DE0C7ABC4F92A0F963C79D60E32A and on 1 October 2021).

**Informed Consent Statement:** Informed consent was obtained from all subjects involved in the study.

**Data Availability Statement:** The data supporting the findings of this study are available at Research Data Unipd at https://doi.org/10.25430/researchdata.cab.unipd.it.00000724 (accessed on 11 October 2023).

**Conflicts of Interest:** The authors report that there are no competing interests to declare.

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
