# Peer review of "Is the Coming Out of an LGBTQIA+ Child a Death-like Event for Italian Parents?"

_socsci, doi:10.3390/socsci12100577_

Round 1

Reviewer 1 Report

The focus of the present study is on parental experiences of their children coming out. The authors use a unique data set from Italy (152 women, 40 men and 2 other gender individuals) to test whether parents’ attitudes and traits are associated with their grief-like experience following their child coming out. For that purpose, the authors administered the Integration of Stressful Life Experiences Scale (ISLES) and the Social Meaning in Life Events Scale (SMILES) to the sample subjects. The authors performed bivariate correlations (or their analogs for ordinal and nominal variables) to probe for a relationship between ISLES, SMILES, and the socio-demographic characteristics of the participants. The authors also compared the sample ISLES scores with scores obtained by Holland et al. (2010) with the bereaved sample originally used to validate the ISLES, using an independent sample t-test. Likewise, they compared the sample SMILES scores with scores obtained by Bellet et al. (2019) with the bereaved sample originally used to validate the SMILES, using the same procedure. No significant differences were found between the sample ISLES and SMILES scores and those validated by their creators, that is by Holland et al. (2010) and Bellet et al. (2019). Moreover, lower ISLES and SMILES scores were associated with higher alexithymia and more traditional gender role beliefs. The authors interpreted their findings as supportive of their original hypothesis that the parental experience of having their children coming out resembles that of bereaved individuals.

The study is both creative and informative, with the goal of deepening our knowledge of the parental experiences following the children coming out. These findings concord with prior evidence describing the emotional experiences of parents who have LGBTQIA+ children:

Saltzburg, S. (2009). Parents' experience of feeling socially supported as adolescents come out as lesbian and gay: A phenomenological study. Journal of Family Social Work, 12(4), 340-358.

Gattamorta, K. A., Salerno, J., & Quidley-Rodriguez, N. (2019). Hispanic parental experiences of learning a child identifies as a sexual minority. Journal of GLBT family studies, 15(2), 151-164.

Still, I have a few questions about the methodology of this study. It is not clear to me why the authors decided to perform Confirmatory Factor Analysis (CFA) on the Gender Roles Beliefs Scale (GRBS). First of all, it seems to me that the authors try to do much in this article. Using the GRBS is a shift of the main focus, and keeping the focus is important for the consistency of the argument. Second, an Italian version of this scale has not yet been validated. Using an English version may be a source of bias. The Italian context may be different from the North American context from which the GRBS was derived. Finally, the authors did not conduct any multivariate analysis from the available data matrix for some reason. Therefore, the relative importance of parental attitudes vis-à-vis other predictors of ISLES and SMILES has not been statistically established.

Further, this study is cross-sectional in nature. This severely limits the authors’ ability to draw causal inferences from their empirical research. As the authors themselves acknowledge, “…parental responses may change and evolve as time passes.” (Line 34). Another important limitation of this study is the possibility of unmeasured confounding variables (e.g., socio-economic status).

Reviewer 2 Report

In general, the article may lead to very interesting results, however, it has strong methodological limitations that need to be clarified before it can be considered for publication:

1. although the theoretical framework is interesting, I would like to see the addition of a section explaining the selection of Italian parents...and why this particular group of parents is relevant? It needs to be clear that the highly Christian-Catholic characteristics of this population group may influence the results and that the results are therefore not generalizable. 

2. I am concerned about the inequality between men and women in the sample. If the majority of the participants are mothers, they tend to have a different perception of their sons, especially if they are the only children or the oldest. In other words, there is a lack of information about the population, since it is not the same what the mother perceives as the father, or what is perceived of a homosexual male son, a lesbian daughter or a transgender son. There are elements that can greatly influence perception and that the study does not make clear.

In this sense, I need a table that points out the characteristics of the participants and their LGBTIQA sons or daughters in order to better understand their response.

3. They talk about the sample being from Northern Italy and 91% from rural areas... this is super important and influences the perception of the participants. This, needs to be pointed out even in the title and from the beginning, because they are Christian Households in Rural Northern Italy. I mean, it's notorious that this is very different than if they were homes in Rome or Barcelona. Therefore, this needs to be well pointed out and the implications made clear.

4. The references need to be updated. Since it is such a current topic, it is necessary that 50% of the references are from the last 5 years.

In general, it is a good article, but it needs to be methodologically clear, otherwise the results are misleading.

Round 2

Reviewer 1 Report

The authors have tried to integrate the feedback from the reviewers to the best of their understanding. The end result aligns with what I have anticipated. 

Author Response

 We would like to thank you for the invaluable help you gave us, allowing us to improve our work and achieve a far better result. We appreciate the time and effort you put into such a detailed review that focused on the core aspects of our research.

Reviewer 2 Report

Many thanks to the authors for the corrections, they have been very valuable.

1. Thank you for table 1, it is very enlightening.

2. Thank you for the improvement in the section on future studies. We know that an article cannot cover all the topics, so it is very important to know that you recognize this and that new lines of work are being considered.

3. Thank you for the additional paragraphs in the discussion.

4. It is still a little low in the number of updated references, however, I recognize the effort.

In general, the author has taken care of what was requested and this is palpable in the article.

Author Response

(The authors gave the same response as above.)
